# MAFFuse: Multi-Attention Fusion Network for Efficient and Robust Image Fusion

Anonymous Full Paper
Submission 63

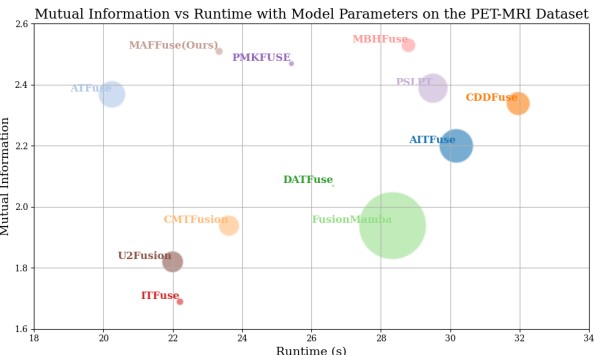

**Figure 1.** Comparison of various image fusion models in terms of mutual information on the Y axis, runtime (test set in seconds) on the X axis, and the number of parameters (in millions) represented by the area of the circle on the PET-MRI dataset. Our model achieves a good balance between performance and computational complexity.

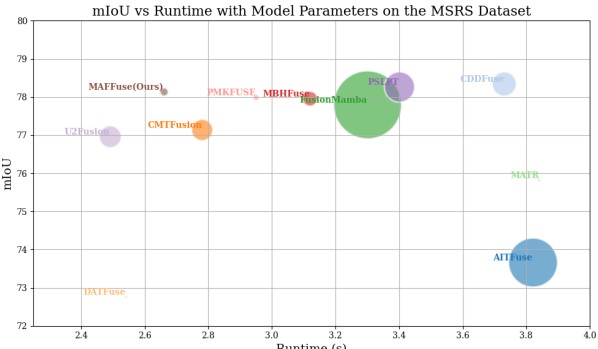

**Figure 2.** Comparison of various image fusion models in terms of mIoU on the Y axis, runtime (per image in seconds) on the X axis, and the number of parameters (in millions) represented by the area of the circle for the downstream semantic segmentation task using the MSRS dataset. Our model achieves a good balance between performance and computational complexity.

## Abstract

Image fusion seeks to combine source images into a single, more informative image while retaining the complementary information from the original images. Existing image fusion models often achieve good results at the cost of increased complexity and computational expense, much of which arises from processing redundant information inherent in strongly correlated images from different sources. In this paper, we introduce an end-to-end lightweight encoder-decoder network that uses channel and spatial attention mechanisms to focus on the most relevant features from multi-source inputs and depthwise convolutions for efficient feature fusion. Our fusion block integrates convolutional layers with a Swin Transformer to capture both local details and global context. Comprehensive evaluations on various benchmarks demonstrate that our approach consistently rivals state-of-the-art methods while maintaining lower computational complexity. Furthermore, we evaluate the fused images on downstream tasks, including semantic segmentation on the MSRS dataset and object detection, showing that our approach enhances task-specific performance. Ablation studies further validate the effectiveness of our specific model design, such as the multi-attention integration, in achieving robust performance with reduced complexity.

## 1 Introduction

Image fusion aims to combine information from multiple sources into a single image, preserving salient features from each input and generating a fused image that is more informative than any individual source. Infrared and Visible Image Fusion (IVF) is particularly relevant for autonomous driving applications, including object detection and semantic segmentation. Medical Image Fusion (MIF), on the other hand, integrates multiple modalities—such as CT-MRI, PET-MRI, and SPECT-MRI—to facilitate faster and more accurate diagnosis. Other fusion techniques, including Multi-Exposure Image Fusion (MEF) and Multi-Focal Image Fusion (MFF), find applications in military operations, such as object tracking and recognition.

Images captured by different sensors exhibit complementary characteristics. Visible images provide rich texture details but can suffer from occlusion under low lighting or adverse weather conditions such as rain and snow. Infrared images, in contrast, capture salient objects effectively under all weather conditions due to their reliance on thermal radiation, though they lack fine textural information. By fusing both modalities, improved performance in tasks like object detection can be achieved even in challenging visual environments. Similar benefits are observed in medical imaging, where MRI delivers

high-resolution anatomical details while PET and SPECT images provide complementary functional information about tissues and organs.

In recent years, deep learning has become the primary driver of advances in image fusion. Existing models are typically categorized into CNN- and Transformer-based approaches. CNNs are widely adopted due to their efficiency in image processing, but their inherently localized receptive fields can limit the capture of long-range dependencies, which are often crucial for multi-modal alignment. Transformer-based approaches, although effective in modeling global context, introduce significant computational overhead, presenting challenges in balancing performance, efficiency, and practical applicability. Furthermore, input images in fusion tasks often exhibit strong semantic correlation, leading to redundancy that can result in unnecessarily complex model designs and the processing of non-discriminative features.

In this work, we show that these challenges can be effectively addressed through improved regulation of *attention* in image fusion models. By integrating lightweight channel and spatial attention mechanisms and incorporating depth-wise convolutions alongside Swin Transformers, we develop a novel image fusion framework that delivers robust performance while significantly reducing computational complexity.

The five main contributions of our paper are as follows:

- We propose MAFFuse, an attention-based CNN and Transformer network for image fusion, which simultaneously captures local and global features while effectively preserving complementary information from source images through improved modeling of global interactions.

- We integrate lightweight channel attention to incorporate global context and spatial attention to suppress irrelevant details, complemented by depthwise convolutions to enhance efficiency without compromising performance.

- Our model achieves performance on par with state-of-the-art methods while reducing computational complexity, as demonstrated in Figure 1 and Figure 2.

- We evaluate the fused images on downstream tasks, including semantic segmentation on the MSRS dataset and object detection, demonstrating that our approach improves task-specific performance.

- We conduct cross-dataset evaluations, training on one dataset and testing on others without additional fine-tuning, highlighting the strong generalization capability of MAFFuse.

## 2 Related Work

Image fusion using deep learning has been addressed with Autoencoders, CNNs, and Transformers, as discussed in the following subsections.

### 2.1 Autoencoder-Based Image Fusion

In image fusion, encoder-decoder architectures are commonly employed for feature extraction and image reconstruction. The encoder extracts key features from input images, which are then processed through a feature fusion block using manually designed strategies such as element-wise addition, averaging, or weighting. An encoder-decoder framework with a fusion block in between was proposed in [8], where the encoder consists of densely connected blocks. This design enhances feature propagation in deep networks and employs the $L1$ norm as a fusion strategy, demonstrating improved fusion effectiveness. A UNet++-style architecture with a fusion strategy that leverages spatial and channel attention modules was later introduced in [10], further improving feature extraction, preserving salient information in the fused image, and retaining fine details from the original images.

### 2.2 CNN-Based Image Fusion

CNN-based image fusion methods often focus on both network architecture and loss function design. A unified CNN-based framework for multi-domain image fusion was proposed in [52], integrating feature extraction, feature fusion, and image reconstruction while employing perceptual loss for training. A DenseNet-based feature extractor was used in [47] to capture fine-grained features and automatically compute the adaptive information preservation of the source images. For multi-modal image fusion without ground truth, an end-to-end self-supervised framework using a UNet-like architecture composed of Transformer and CNN modules was introduced in [54] to process cross-modal features effectively.

Attention mechanisms aim to replicate human visual attention by focusing on salient regions in an image and adaptively weighting features according to their importance. Incorporating channel-wise attention into a CNN allows the network to capture global dependencies, mitigating the limitations of local receptive fields [6]. Spatial attention mechanisms have been applied to improve long-range dependency modeling in CNNs [41], while the CBAM module [45] combines separate channel and spatial attention to enhance feature representations. To improve efficiency without sacrificing performance, a lightweight channel attention mechanism using 1D convolution was proposed in [39].

## 2.3 Transformer-Based Image Fusion

CNNs incorporate strong image-specific inductive biases, but their limited receptive fields make it challenging to model long-range dependencies. To address this limitation, transformer-based methods have been adopted. Initially proposed to capture long-range dependencies in natural language processing [37], transformers were later adapted for computer vision by dividing images into patches before serialization [4]. Leveraging this capability for image fusion, a Y-shaped end-to-end transformer network was introduced in [31], featuring two parallel branches to separately extract textural details from visible images and thermal radiation information from infrared images. Similarly, [32] proposed an adaptive integration of convolution and transformer modules for multimodal medical image fusion, employing a multi-scale network to capture features at various scales and enhance global context modeling. Multi-scale transformer branches were employed to aggregate features while preserving modality-specific information [30].

Further developments include the integration of intra-domain self-attention with inter-domain cross-attention for efficient feature extraction and fusion within and across domains [16], and the use of a gated bottleneck with cross-attention to remove redundant spatial and channel information while preserving complementary source features [19]. Dual-attention-residual transformer modules have been employed to extract the most salient features [34]. Several works have explored dual-branch CNN-transformer networks for more effective fusion. For instance, a correlation-based loss was used to suppress noise in extracted features [53]. Additionally, [40] proposed a model that decomposes source images into multi-frequency features and applies learned fusion rules using Swin Transformers, with an architecture comprising shared encoders, a fusion block, and a decoder. MixFuse [11] combines self-attention and cross-attention transformer modules to enhance modality-specific feature extraction, compute inter-feature correlations, and remove redundant information.

# 3 Methodology

## 3.1 Problem Definition

In this paper, we consider bi-modality image fusion. For MFF and MEF, both input images are in RGB ($I \in R^{H \times W \times 3}$) format. In cases of IVF and MIF, one image is in RGB ($I_1 \in R^{H \times W \times 3}$) format, while the other is in grayscale ($I_2 \in R^{H \times W \times 1}$) format. The fusion objective is to combine the original input images into a single RGB image ($I_f \in R^{H \times W \times 3}$) that preserves key information from the inputs. For IVF and MIF tasks, where a 3-channel image is fused with a 1-channel image, we address the channel mismatch by converting the RGB image to YUV and extracting its Y, U, and V components. The Y component of the YUV image is fed along with the grayscale image into our network. Since our network is trained end-to-end, it avoids the need for manually designed fusion strategies. Finally, the fused output is converted from YUV back to RGB. In the case of the Oocytes dataset, 11 images are in GrayScale ($I_1 \in R^{H \times W \times 1}$) format. The objective is to combine the 11 images into a single gray-scale fused image ($I_f \in R^{H \times W \times 1}$).

## 3.2 Model Architecture

### 3.2.1 Attention Module

To further enhance attention modeling beyond existing approaches in image fusion, we adopt the Efficient Channel Attention (ECA) structure [39], where the main motivation is to replace coarse dimensionality reduction with granular local cross-channel interaction. We follow the three-layer design where Global Average Pooling (GAP) is first used, which generates a channel descriptor, followed by a 1D convolutional layer with a kernel size of 7, and finally, a sigmoid activation that produces the attention map. Specifically, input features, $F \in R^{W \times H \times C}$, are processed with GAP to yield a feature $F_{gap} \in R^{1 \times 1 \times C}$, which is then passed through a 1D convolutional layer and activated by a sigmoid function. This process is summarized as Equation 1:

$$M_c(F) = \sigma \left( \text{Conv1D} \left( \frac{1}{H \cdot W} \sum_{h=1}^{H} \sum_{w=1}^{W} F_{hw} \right) \right), \tag{1}$$

Where $\sigma$ is the sigmoid activation function. The inner term is the GAP operation, $F_{gap}$. To better capture (localized) saliency in feature maps, we enhance our attention design by integrating the Spatial Attention (SA) module [45], which weights feature maps based on their importance. The SA module comprises a convolutional layer ($\text{Conv}^{7 \times 7}$), a sigmoid activation function ($\sigma$), and (max and average) pooling operations. The intermediate feature maps are concatenated and convolved to obtain the final feature map $M_s \in R^{1 \times H \times W}$ as shown in Equation 2:

$$M_s(F) = \sigma \left( \text{Conv}^{7 \times 7} \left( \text{Concat} \left[ F_{\text{Avg}}^S, F_{\text{Max}}^S \right] \right) \right) \tag{2}$$

Where $M_s(F)$ represents the output feature map, $F_{Avg}$ and $F_{Max}$ represent the average and max pooling operations along the channel axis, respectively. Assuming an input feature map $F$, the final (channel and spatial) attention-enhanced feature map is computed by element-wise multiplying $F$ with both

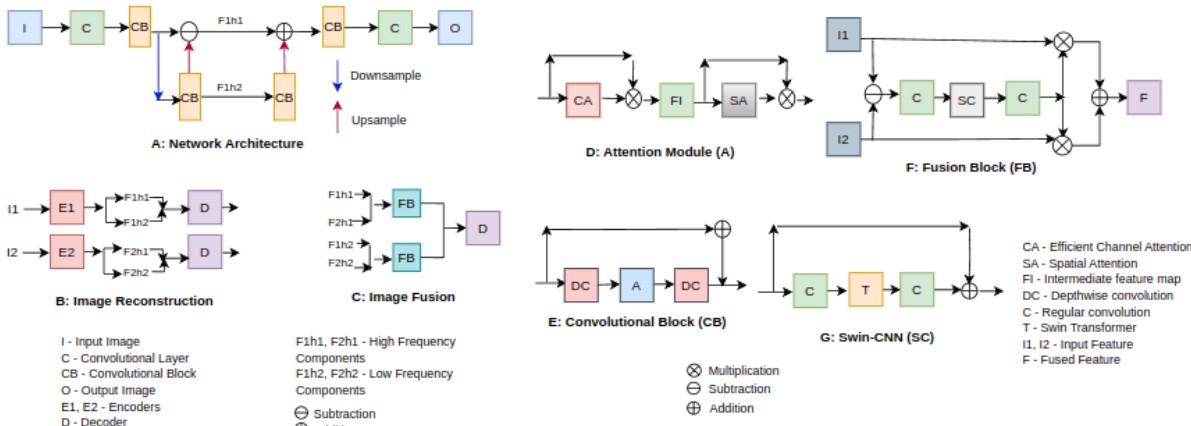

**Figure 3.** A: Overview of the proposed network architecture. B: Image reconstruction process within the network. C: Image fusion procedure. D: Proposed attention module. E: Attention-based convolutional block. F: Fusion block. G: Swin-CNN hybrid block.

the channel and spatial attention maps, as shown in Equation 3 and Equation 4 [45]. We note that instead of combining CA and SA as demonstrated in [45], our attention design is motivated by integrating ECA and SA.

$$F_c = M_c\left(F\right) \otimes F, \qquad (3)$$

$$F_o = M_s\left(F_c\right) \otimes F_c. \qquad (4)$$

Here $\otimes$ denotes element-wise multiplication and $F_o$ is the final attention-enhanced feature map. In our fusion model, the proposed attention design Figure 3 (D) is integrated between two consecutive convolutional layers via a residual connection, as depicted in Figure 3 (E).

### 3.2.2 Encoder-Decoder and Fusion Block

To construct a complete encoder-decoder framework, we adopt an architecture inspired by [40]. This design includes two encoder blocks for the source images, a fusion block, and a shared decoder. Our complete framework is depicted in Figure 3 (A) with Image Reconstruction and Image Fusion depicted in Figure 3 (B) and Figure 3 (C) respectively.

In our framework, each source image is independently processed through an encoder to extract features, where the residual between (multi-frequency) features is computed (shown as the − symbol in Figure 3) (A) and fed into a fusion block. As depicted in Figure 3 (C), the fusion block [40] mainly comprises two convolutional layers with a Swin Transformer in between Figure 3 (F). The advantage of this design is that, unlike hand-crafted fusion techniques, it enables a more flexible combination of local and global features at different frequencies from the source images. After applying a softmax operation, the resulting learned features, $F_1$ and $F_2$

are element-wise multiplied with the source image features $I_1$ and $I_2$, respectively, to compute the final fused feature, $F_{out}$ is defined as in Equation 5:

$$F_{out} = I_1 \odot F_1 + I_2 \odot F_2 \qquad (5)$$

where $\odot$ denotes the elementwise multiplication.

Compared to the original design in [40], where four convolutional and Swin transformer-based blocks are employed in the encoder and decoder, our proposed model follows a more efficient design using two attention-based depth-wise [3] convolutional blocks without a transformer. Indeed, we trade complexity with enhanced attention regulation by incorporating an attention module, as shown in Figure 3 (D), between the two depthwise convolutional layers within our block Figure 3 (E).

## 3.3 Loss Function

To enhance our fusion performance, we employ an ensemble loss that combines several loss functions during training, in an unsupervised manner.

Structural similarity (SSIM) loss is included in our ensemble to preserve the source images' structural information. This loss quantifies the similarity between the fused image and the source images in terms of brightness, contrast, and structure. The SSIM loss is defined in Equation 6 and Equation 7 as:

$$L_{\text{ssim}} = 1 - \text{SSIM}\left(I_Y, I_X\right), \qquad (6)$$

$$\text{SSIM}(X,Y) = \frac{(2\mu_X\mu_Y + C_1)(2\sigma_{XY} + C_2)}{(\mu_X{}^2 + \mu_Y{}^2 + C_1)(\sigma_X^2 + \sigma_Y^2 + C_2)}, \qquad (7)$$

where $X$ and $Y$ represent the source and fused images, respectively. $(\mu_X, \sigma_X)$ denote the mean and standard deviation of $X$ and $(\mu_Y, \sigma_Y)$ denote the

mean and standard deviation of $Y$. The correlation between $X$ and $Y$ is represented as $\sigma_{(}X, Y)$. $C_1 = 0.01$ and $C_2 = 0.03$ are set as constants in our experiments.

Gradient-based loss is included to retain edge saliencies from the two source images. The gradient loss, computed using the Sobel operator, is defined as Equation 8:

$$L_{\text{gradient}} = \frac{1}{HW} \| |\nabla I_f| - \max(|\nabla I_{inp1}|, |\nabla I_{inp2}|) \|_1 . \tag{8}$$

where $\nabla$ denotes the gradient operator that measures the texture information in an image and that is computed using the Sobel operation. $H$ and $W$ represent the height and width of the two source images ($I_{inp1}$, $I_{inp2}$), respectively. $I_f$ represents the fused image.

The intensity loss is included to regulate the pixel intensity distribution of the fused image concerning the two source images. The intensity loss is defined in Equation 9 as:

$$L_{int} = \frac{1}{HW} \| I_f - \max(I_{inp1}, I_{inp2}) \|_1 , \tag{9}$$

The final loss ensemble $L_{total}$ is therefore composed as Equation 10 with 50, 100, and 20 as the weights chosen for the SSIM, Gradient, and Intensity loss:

$$L_{total} = 50 * L_{SSIM} + 100 * L_{Gradient} + 20 * L_{Intensity} \tag{10}$$

# 4 Experiment

## 4.1 Datasets

We evaluate our model on multiple benchmark datasets and compare it with state-of-the-art image fusion methods. The following datasets were considered in our experiments:

1. **PET-MRI (MIF Task):** Sourced from the Harvard Medical Dataset, this dataset contains 311 PET-MRI image pairs, split into 269 for training and 42 for testing with a resolution of $256 \times 256$ [23].

2. **MSRS (IVF Task):** This dataset comprises 1444 pairs of urban infrared and visible images, with 1083 pairs allocated for training and 361 for testing. Each image has a resolution of $480 \times 640$ [28].

3. **MFI-WHU (MFF Task):** This dataset consists of 120 near-focused and far-focused image pairs. The dataset is used only for training. The images vary in resolution [49].

4. **Lytro (MFF Task):** This dataset consists of 20 near-focused and far-focused image pairs. The dataset is used only for testing. The images are of resolution $520 \times 520$. [18]

5. **SICE (MEF Task):** This dataset consists of 589 over-exposed and under-exposed image brightness pairs. The dataset is used only for training. The images vary in resolution. [1]

6. **MEFB (MEF Task):** Comprising 99 image pairs, this dataset is exclusively used for testing. The images have varying resolutions [50].

7. **AWMM-100k (IVF Task):** The large-scale benchmark of 20 image pairs across diverse weather and lighting conditions of various resolutions is used for testing [12].

8. **Oocytes (MFF Task):** This private dataset contains 2167 multi-focal (11 focal) images, split into 1606 images for training and 561 images for testing with a resolution of $800 \times 800$.

## 4.2 Implementation Details

Training is conducted using the Adam optimizer with an initial learning rate of $10^{-4}$, which is adjusted using the MultiStepLR scheduler by reducing the learning rate by a factor of 0.5 every 50 epochs. We use a batch size of 4, and our training is restricted to 200 epochs on an Nvidia A100 GPU. The images are randomly cropped into patches of size $64 \times 64$ for training.

## 4.3 Evaluation Metrics

The performance of our model is evaluated using various metrics, each reflecting a different aspect of fusion quality. Specifically, Entropy (EN) measures the amount of unique information, Standard Deviation (SD) reflects color variation, Spatial Frequency (SF) evaluates edge details, and Average Gradient (AG) assesses edge sharpness. Additionally, Mutual Information (MI) indicates how much information is retained from the original images, and the Structural Content Difference (SCD) assesses information preservation. Higher values are preferred for all metrics. We note that EN, SD, SF, and AG are no-reference metrics that do not require a ground-truth image, whereas MI and SCD are reference-based, comparing the fused image with the source images. For evaluating the computational complexity of our model, we use the number of parameters in the model (Params) in millions, floating point operations per second in the model (FLOPS) in GigaFLOPS, and the time to inference on the test set (Runtime) in seconds. The FLOPS is calculated on images with a resolution of $256 \times 256$. A lower value is preferred for Params, FLOPS, and Runtime.

## 4.4 Comparison Approaches

We compare our model with other state-of-the-art image fusion methods, including AITFuse [44], ATFuse [7], CDDFuse [53], CMTFusion [19], DATFuse [34], FusionMamba [46] ITFuse [35], MATR [32], MBHFuse [24], PMKFuse [25], PSLPT [40], U2Fusion [47], and YDTR [31]. The performance of these methods is reproduced using open-sourced implementations provided by the original authors, following similar experimental settings described in their respective papers. For the model proposed in [40], we train it using an unsupervised manner instead of the semi-supervised manner used by the authors to maintain consistency with all other approaches and ours.

### 4.4.1 Quantitative Performance

Quantitative performance comparisons are carried out on benchmark datasets including PET-MRI and MSRS, as shown in Table 1 (A and B). Our model consistently performs competitively with state-of-the-art methods while substantially reducing computational complexity compared to most approaches.

To further investigate the generalization ability of our model, we evaluate the trained model on the MFI-WHU dataset (MFF task) and on the Lytro dataset (MFF task) without any re-training or fine-tuning. Similarly, for the MEF task, we train on the SICE dataset and test on the MEFB dataset. Quantitative comparisons with other methods under these two cross-validation scenarios are presented in Table 1 (C and D), where our model again demonstrates competitive performance. We note that our approach performs better on some datasets (eg, Lytro) than on others (eg, PET-MRI) and believe that the reason behind this is the blurred, low-SNR inputs in the case of the PET-MRI dataset.

Our method ranks second using MI as the metric on both the PET-MRI and Lytro datasets. While on the MSRS dataset, our method ranks second using SD as the metric. Our network architecture has lower Params and FLOPS than all models except DATFuse, ITFuse, MATR, and PMKFuse. We note that the models with better performance (AITFuse, CDDFuse, CMTFusion, and FusionMamba) using fusion quality metrics have a higher number of parameters. Our model achieves a good tradeoff between performance and efficiency.

### 4.4.2 Qualitative Performance

The qualitative performance of our model compared to other methods is shown in Figure 4 and Figure 5 using the PET-MRI and MSRS benchmarks, respectively.

DATFuse and FusionMamba lead to excessive sharpening in the fused image, while CMTFusion, ITFuse, and U2Fusion lead to blurry details in the fused image. CMTFusion and U2Fusion produce noisy results, while CMTFusion, ITfuse, and FusionMamba result in artifacts in the fused image to varying degrees. Our method results in fused images with less blur, noise, distortion, and artifacts than most other methods. Compared with methods like AITFuse, CDDFuse, MBHFuse, and PMKFuse, our method results in a fused image with higher contrast and clearer details. AITFuse, CDDFuse, CMTFusion, ITFuse, MBHFuse, PMKFuse, and our method all undergo color distortion to varying extents.

The qualitative comparison of our approach with existing state-of-the-art methods on the Oocytes

| Method | EN ↑ | SD ↑ | SF ↑ | AG ↑ | MI ↑ | SCD ↑ | Param ↓ | FLOPS ↓ | Time ↓ |
|---|---|---|---|---|---|---|---|---|---|
| **A: PET-MRI Dataset** | | | | | | | | | |
| AIT [44] | **6.41** | 55.78 | 8.41 | **3.48** | 2.20 | 1.24 | 6.50 | 82.09 | 30.16 |
| ATF [7] | 4.01 | 71.41 | 8.22 | 2.86 | 2.37 | 1.22 | 1.05 | 5.40 | **20.24** |
| CDD [53] | 4.07 | 61.01 | 8.10 | 2.77 | 2.34 | **1.29** | 1.19 | 77.68 | 31.95 |
| CMT [19] | 4.04 | 47.64 | 6.73 | 2.40 | 1.94 | 1.19 | 0.62 | 13.10 | 23.61 |
| DAT [34] | 4.18 | **87.38** | 7.45 | 2.53 | 2.07 | 1.19 | **0.01** | **2.32** | 26.62 |
| FMB [46] | 4.30 | 67.80 | **8.73** | 3.06 | 1.94 | 1.20 | 225.42 | 26.48 | 28.32 |
| ITF [35] | 4.34 | 38.53 | 5.82 | 1.97 | 1.69 | 1.23 | 0.08 | 5.68 | 22.20 |
| MBH [24] | 3.93 | 60.26 | 8.11 | 2.78 | **2.53** | 1.19 | 0.30 | 28.92 | 28.78 |
| PMK [25] | 3.91 | 61.42 | 8.10 | 2.75 | 2.47 | 1.28 | 0.05 | 3.29 | 25.41 |
| PSL [40] | 4.06 | 66.94 | 8.07 | 2.79 | 2.39 | 1.26 | 1.26 | 24.56 | 29.50 |
| U2F [47] | 4.49 | 47.00 | 5.28 | 2.02 | 1.82 | 1.20 | 0.66 | 43.17 | 21.98 |
| Ours | 3.94 | 66.32 | 8.01 | 2.81 | 2.51 | 1.25 | 0.09 | 4.95 | 23.33 |
| **B: MSRS Dataset** | | | | | | | | | |
| AIT [44] | 5.75 | 39.71 | 4.83 | 1.97 | 2.98 | 0.99 | 6.50 | 82.09 | 1377.64 |
| CDD [53] | 6.69 | **43.40** | 5.80 | 2.60 | 3.68 | 1.06 | 1.19 | 77.68 | 1347.74 |
| CMT [19] | 5.97 | 23.74 | 4.61 | 1.95 | 2.46 | 1.18 | 0.62 | 13.10 | 1002.73 |
| DAT [34] | 6.40 | 36.22 | 4.93 | 2.25 | 2.62 | 1.05 | 0.01 | 2.32 | 918.12 |
| FMB [46] | 6.53 | 38.29 | **7.39** | **3.01** | 2.48 | 1.12 | 225.42 | 26.48 | 1163.63 |
| ITF [35] | 4.08 | 5.61 | 1.90 | 0.81 | 1.42 | 0.20 | 0.08 | 5.68 | 973.03 |
| MATR [32] | 5.92 | 24.58 | 4.84 | 2.08 | 2.49 | **1.17** | 0.01 | 3.90 | 1385.34 |
| MBH [24] | 6.68 | 42.63 | 6.06 | 2.73 | **3.74** | 1.06 | 0.30 | 28.92 | 1125.11 |
| PMK [25] | **6.73** | 43.24 | 6.13 | 2.78 | 3.53 | 1.05 | 0.05 | 3.29 | 1117.28 |
| PSL [40] | 6.50 | 42.34 | 5.59 | 2.47 | 2.97 | 1.07 | 1.26 | 24.56 | 1209.86 |
| U2F [47] | 5.92 | 24.27 | 3.68 | 1.53 | 2.55 | 1.16 | 0.66 | 43.17 | **899.58** |
| YDTR [31] | 4.31 | 6.34 | 2.54 | 1.01 | 1.47 | 1.05 | 0.22 | 20.58 | 996.23 |
| Ours | 6.55 | 43.33 | 5.66 | 2.53 | 3.06 | 1.06 | 0.09 | 4.95 | 960.75 |
| **C: Lytro Dataset** | | | | | | | | | |
| AIT [44] | 7.55 | 59.44 | 8.19 | 3.87 | 6.71 | 1.11 | 6.50 | 82.09 | 94.59 |
| ATF [7] | 7.46 | 55.77 | 8.09 | 3.72 | 6.95 | 1.11 | 1.05 | 5.40 | 45.98 |
| CDD [53] | 7.57 | 58.79 | 8.15 | 3.82 | 7.19 | **1.12** | 1.19 | 77.68 | 65.01 |
| CMT [19] | 7.53 | 53.40 | 6.31 | 3.10 | 6.34 | 1.11 | 0.62 | 13.10 | 46.17 |
| DAT [34] | 7.21 | 62.09 | 7.21 | 3.19 | 6.94 | 1.12 | 0.01 | 2.32 | **44.27** |
| FMB [46] | 7.55 | **71.82** | **10.86** | **5.17** | **10.02** | 0.83 | 225.42 | 26.48 | 55.11 |
| ITF [35] | 5.93 | 21.99 | 3.49 | 1.47 | 5.48 | 1.03 | 0.08 | 5.68 | 46.06 |
| MATR [32] | 6.97 | 39.45 | 6.96 | 3.06 | 6.32 | 1.11 | 0.01 | 3.90 | 61.46 |
| MBH [24] | 7.52 | 56.91 | 8.01 | 3.72 | 7.22 | 1.12 | 0.30 | 28.92 | 60.17 |
| PMK [25] | **7.58** | 60.11 | 8.88 | 4.26 | 6.99 | 1.11 | 0.05 | 3.29 | 59.87 |
| PSL [40] | 7.52 | 56.70 | 7.83 | 3.67 | 7.13 | 1.12 | 1.26 | 24.56 | 63.26 |
| U2F [47] | 6.33 | 23.88 | 2.84 | 1.27 | 6.10 | 1.09 | 0.66 | 43.17 | 46.68 |
| YDTR [31] | 7.24 | 48.83 | 7.20 | 3.14 | 6.48 | 1.11 | 0.22 | 20.58 | 50.78 |
| Ours | 7.53 | 57.23 | 7.85 | 3.67 | 7.24 | 1.11 | 0.09 | 4.95 | 50.15 |
| **D: MEFB Dataset** | | | | | | | | | |
| AIT [44] | 4.55 | 39.57 | 5.08 | 1.72 | 3.60 | 1.17 | 6.50 | 82.09 | 388.39 |
| ATF [7] | 6.57 | 63.82 | 7.94 | 2.88 | 4.67 | 1.49 | 1.05 | 5.40 | **321.13** |
| CDD | 7.00 | 65.50 | 7.64 | 3.29 | 5.45 | 1.50 | 1.19 | 77.68 | 387.61 |
| CMT [19] | 6.88 | 51.07 | 6.01 | 2.50 | 4.36 | **1.57** | 0.62 | 13.10 | 366.73 |
| DAT [34] | 7.10 | **70.30** | 6.99 | 2.72 | 4.48 | 1.45 | **0.01** | **2.32** | 357.11 |
| FMB [46] | 6.72 | 63.93 | **9.21** | **4.00** | **7.14** | 1.20 | 225.42 | 26.48 | 425.03 |
| ITF [35] | 6.47 | 65.67 | 5.91 | 2.29 | 5.03 | 1.48 | 0.08 | 5.68 | 379.37 |
| MATR [32] | 5.69 | 20.29 | 7.77 | 2.56 | 3.63 | 1.44 | 0.01 | 3.90 | 469.17 |
| MBH [24] | 6.80 | 60.44 | 7.49 | 3.19 | 5.52 | 1.48 | 0.30 | 28.92 | 380.76 |
| PMK [25] | **7.18** | 69.01 | 7.84 | 3.36 | 5.57 | 1.53 | 0.05 | 3.29 | 387.66 |
| PSL [40] | 6.56 | 58.85 | 7.18 | 2.97 | 4.73 | 1.50 | 1.26 | 24.56 | 382.06 |
| U2F [47] | 6.58 | 38.67 | 3.38 | 1.42 | 4.86 | 1.54 | 0.66 | 43.17 | 366.19 |
| YDTR [31] | 6.29 | 51.82 | 7.07 | 2.73 | 4.17 | 1.52 | 0.22 | 20.58 | 378.86 |
| Ours | 6.45 | 60.62 | 6.93 | 2.82 | 5.54 | 1.49 | 0.09 | 4.95 | 359.65 |
| **E: Oocytes Dataset** | | | | | | | | | |
| EMMA [54] | 7.21 | 64.61 | 3.02 | 1.28 | 15.42 | 10.23 | 1.34 | 28.42 | 11267.04 |
| PMK [25] | 7.27 | 45.42 | 3.51 | 1.47 | 12.64 | 9.12 | 0.31 | 22.09 | 11709.42 |
| PSL [40] | 7.22 | 65.70 | 3.12 | 1.31 | 15.27 | 10.20 | 4.79 | 84.25 | 11542.88 |
| U2F [47] | 7.19 | 64.92 | 2.92 | 1.24 | 15.47 | 10.23 | 0.66 | 43.40 | 11139.84 |
| Ours | 7.15 | 63.83 | 2.99 | 1.24 | 15.51 | 10.22 | 0.12 | 6.58 | 11203.21 |

**Table 1.** Quantitative comparison across multiple datasets and tasks. A: PET-MRI dataset (MIF task). B: MSRS dataset (IVF task). C: Lytro dataset (MFF task) evaluated using the model trained on MFI-WHU without fine-tuning. D: MEFB dataset (MEF task) evaluated using the model trained on SICE without fine-tuning. E: Oocytes dataset (MFF task). The best-performing values are highlighted in bold.

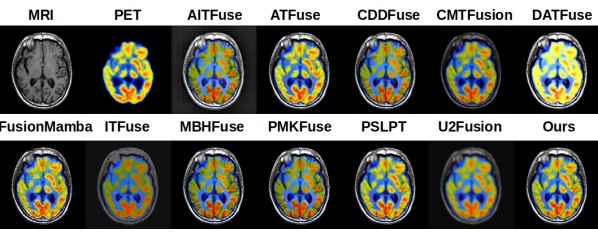

**Figure 4.** Qualitative comparison on the PET-MRI dataset (MIF Task) with other state-of-the-art methods.

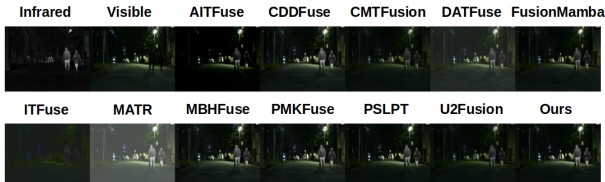

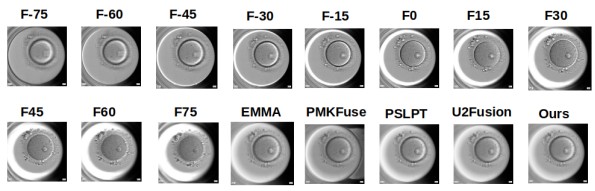

**Figure 5.** Qualitative comparison on the MSRS dataset (IVF Task) with other state-of-the-art methods.

| Method | Background | Car | Person | Bike | Curve | Car Stop | Guardrail | Color Tone | Bump | mIoU |
|---|---|---|---|---|---|---|---|---|---|---|
| IR | 96.14 | 61.90 | 70.00 | 24.46 | 33.64 | 20.66 | 0.06 | 20.98 | 27.97 | 39.53 |
| VI | 97.92 | 86.79 | 39.97 | 70.50 | 53.33 | 71.84 | **85.90** | 65.44 | 79.16 | 72.32 |
| AIT | 98.25 | 87.68 | 70.39 | 67.76 | 51.25 | 68.89 | 83.03 | 61.06 | 74.71 | 73.67 |
| CDD | **98.61** | 90.42 | 74.52 | **72.16** | 65.22 | 74.39 | 84.42 | 65.95 | 79.44 | **78.35** |
| CMT | 98.56 | 90.40 | 74.19 | 71.08 | 62.03 | 75.54 | 81.31 | 65.42 | 75.72 | 77.14 |
| DAT | 98.27 | 88.58 | 72.06 | 66.90 | 53.11 | 66.69 | 82.03 | 61.27 | 66.0 | 72.77 |
| FMB | 98.58 | 90.36 | 73.85 | 71.75 | 63.43 | **75.85** | 82.79 | 65.70 | 77.89 | 77.80 |
| ITF | 96.41 | 65.80 | 60.66 | 51.13 | 19.17 | 39.76 | 37.87 | 48.62 | 8.54 | 47.55 |
| MATR | 98.47 | 89.59 | 73.97 | 69.98 | 60.15 | 70.76 | 83.38 | 64.15 | 72.04 | 75.83 |
| MBH | 98.60 | **90.49** | **74.68** | 71.91 | **65.42** | 74.43 | 84.55 | 65.72 | 75.81 | 77.96 |
| PMK | 98.58 | 90.29 | 74.40 | 72.00 | 64.97 | 74.87 | 84.49 | **66.22** | 76.09 | 77.99 |
| PSL | 98.60 | 90.37 | 74.50 | 71.85 | 64.46 | 74.16 | 84.83 | 66.10 | **79.61** | 78.27 |
| U2F | 98.50 | 89.80 | 73.32 | 70.44 | 62.17 | 74.29 | 82.00 | 64.77 | 77.46 | 76.97 |
| YDTR | 96.80 | 70.36 | 56.65 | 57.27 | 28.04 | 60.14 | 54.86 | 54.59 | 14.55 | 54.81 |
| Ours | 98.59 | 90.36 | 74.55 | 71.80 | 64.46 | 74.10 | 84.87 | 66.02 | 78.52 | 78.14 |

**Table 2.** Quantitative comparison on the MSRS dataset with state-of-the-art methods for the downstream semantic segmentation task using the BiSeNetV2 model. The best-performing values are highlighted in bold.

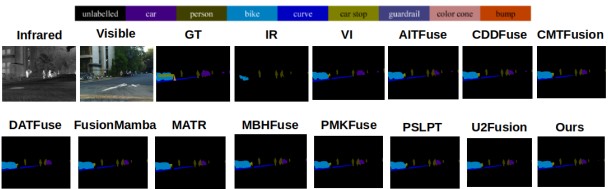

**Figure 8.** Qualitative comparison on the MSRS dataset with state-of-the-art methods for the downstream semantic segmentation task using the BiSeNetV2 model. GT denotes the ground truth, while IR and VI represent the segmentation maps obtained from the infrared and visible images, respectively.

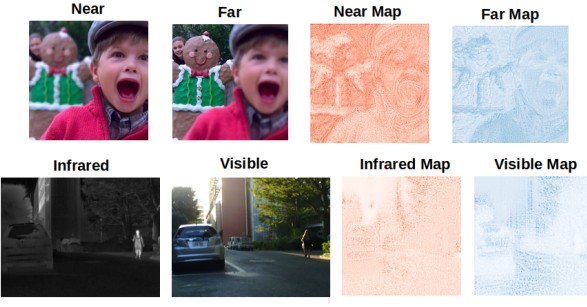

**Figure 6.** Qualitative comparison on the Oocytes dataset with other state-of-the-art methods. Here, F0, F15, F30, F45, F60, F75, F-15, F-30, F-45, F-60, and F-75 denote 11 different focal lengths.

dataset is presented in Figure 6. The results demonstrate that our method achieves performance comparable to, and in some cases surpassing, current state-of-the-art techniques.

The importance map visualization, highlighting the relative contributions of the two source images to the fused output, is shown in Figure 7.

**Figure 7.** Importance map visualization. The top row, from left to right, shows a near-focus image, a far-focus image, the corresponding near importance map, and the far importance map using a sample from the Lytro dataset. The bottom row, from left to right, displays an infrared image, a visible image, the corresponding infrared importance map, and the visible importance map using a sample from the MSRS dataset.

### 4.4.3 Application of Downstream Tasks

We evaluate the fused images on the downstream task of semantic segmentation using the MSRS dataset and observe that our approach achieves the third-highest score in terms of the mIoU metric, as reported in Table 2. A qualitative comparison of the resulting segmentation maps is shown in Figure 8.

### 4.4.4 Ablation Study

We conduct ablation experiments using the PET-MRI benchmark to investigate the key architectural design choices of our proposed image fusion model, focusing on our attention module (ECA+SA) and the architectural decision to trade model complexity for more effective attention regulation by optimal usage of Swin Transformer supplemented by our enhanced attention module in Table 3. Furthermore, an ablation study using regular and depthwise convolution is shown in Table 4. Finally, a loss function-based ablation study is depicted in Table 5.

Firstly, as shown in Table 3(A), combining ECA and SA in our attention module consistently delivers the best performance across most of the evaluation metrics. ECA enhances channel-aware detail selection, improving fusion clarity, while SA complements this by focusing on local spatial importance. Together, they improve the fidelity and sharpness of fused outputs. In particular, ECA enhances local contrast and discriminability of modality-specific features (e.g., structures in PET and edges in infrared). On the other hand, SA acts as an adaptive filter to reinforce key local spatial regions. ECA alone improves channel selectivity and detail enhancement but fails to capture global context. SA alone improves spatial localization but introduces background noise without channel filtering. While using the Swin transformer alone makes the fused image sensitive to modality misalignment.

Secondly, including a transformer only in the fusion block allows the model to capture correlations between input images without attending to redun-

| SSIM | Gradient | Intensity | EN ↑ | SD ↑ | SF ↑ | AG ↑ | MI ↑ | SCD ↑ |
|---|---|---|---|---|---|---|---|---|
| 10 | 50 | 100 | 3.60 | 64.50 | 7.80 | 2.70 | 2.45 | 1.20 |
| 30 | 50 | 100 | 3.85 | 65.00 | 7.95 | 2.75 | 2.48 | 1.22 |
| 20 | 30 | 100 | 3.65 | 64.60 | 7.82 | 2.72 | 2.46 | 1.21 |
| 20 | 70 | 100 | **3.98** | 65.10 | 7.98 | 2.78 | 2.50 | 1.23 |
| 20 | 50 | 50 | 3.55 | 64.30 | 7.75 | 2.68 | 2.44 | 1.19 |
| 20 | 50 | 200 | 3.92 | 66.00 | 8.00 | **2.80** | 2.49 | 1.24 |
| 20 | 50 | 100 | 3.94 | **66.32** | **8.01** | 2.81 | **2.51** | **1.25** |

**Table 5.** Quantitative ablation study on the PET-MRI dataset evaluating different combinations of SSIM, Intensity, and Gradient loss terms. The best-performing values are highlighted in bold.

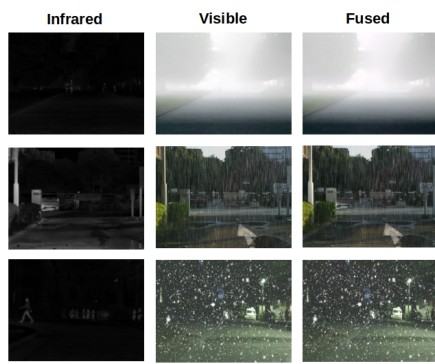

**Figure 9.** Some of the failure cases on the AWMM-100k dataset using our approach. A human is seen in each of the infrared scenes but not in the corresponding visible images. The fused images miss the human completely.

dant information within each source image. As shown in Table 3 (B), this approach provides the best balance between performance and complexity, yielding a clear improvement over either using transformers in all blocks (encoder, decoder, and fusion) or removing transformers entirely from our model design or using a transformer only in the encoder and decoder blocks. Using the Swin transformer only in the fusion block produces the best results in terms of contrast and modality-specific details. The role of Swin Transformer is to encode both long-range structural dependencies and multi-scale context.

ECA and SA attention mechanisms are integrated into the convolutional blocks to suppress redundant information and enhance the extraction of salient features. Additionally, the Swin Transformer is incorporated exclusively in the fusion block, allowing the model to capture global correlations between input images while minimizing redundant information.

Thirdly, as shown in Table 4, employing depthwise convolutions in the encoder and decoder offers a performance boost over standard convolutions on multiple metrics while significantly reducing model complexity.

| A: Attention Module | | | | | | | | | | |
|---|---|---|---|---|---|---|---|---|---|---|
| SA | ECA | EN ↑ | SD ↑ | SF ↑ | AG ↑ | MI ↑ | SCD ↑ | Param ↓ | FLOPS ↓ | Time ↓ |
| × | × | 3.96 | 67.93 | **8.08** | 2.82 | 2.51 | 1.24 | **0.09** | **4.92** | 10.03 |
| × | ✓ | **3.97** | **68.70** | 8.06 | **2.82** | 2.51 | 1.24 | 0.09 | 4.93 | 14.38 |
| ✓ | × | 2.30 | 8.21 | 1.89 | 0.51 | 1.71 | 1.15 | 0.09 | 4.94 | 19.54 |
| ✓ | ✓ | 3.94 | 66.32 | 8.01 | 2.81 | **2.51** | 10.15 | 0.09 | 4.95 | 23.33 |
| B: Transformer | | | | | | | | | | |
| Enc/Dec | Fusion | EN ↑ | SD ↑ | SF ↑ | AG ↑ | MI ↑ | SCD ↑ | Param ↓ | FLOPS ↓ | Time ↓ |
| × | × | **3.97** | **67.24** | 7.95 | 2.77 | 2.51 | 0.03 | 0.37 | **2.21** | 14.53 |
| ✓ | × | 2.88 | 59.84 | 6.08 | 1.71 | 1.80 | 0.97 | 0.37 | 10.45 | 21.09 |
| ✓ | ✓ | 3.95 | 62.55 | **8.05** | 2.76 | 2.29 | **1.27** | 0.42 | 13.20 | 27.85 |
| × | ✓ | 3.94 | 66.32 | 8.01 | **2.81** | **2.51** | 1.25 | 0.09 | 4.95 | 23.33 |

**Table 3.** A: Quantitative ablation study on the PET-MRI dataset evaluating the attention module, where SA denotes Spatial Attention and ECA denotes Efficient Channel Attention. B: Quantitative ablation study on the PET-MRI dataset evaluating the Swin Transformer block, where Enc/Dec refers to the encoder-decoder block and Fusion refers to the fusion block. The best-performing values are highlighted in bold.

| Conv | EN ↑ | SD ↑ | SF ↑ | AG ↑ | MI ↑ | SCD ↑ | Param ↓ | FLOPS ↓ | Time ↓ |
|---|---|---|---|---|---|---|---|---|---|
| Reg | **3.96** | **68.43** | 8.01 | **2.82** | 2.50 | 1.24 | 0.33 | 14.93 | 28.83 |
| D/w | 3.94 | 66.32 | **8.01** | 2.81 | **2.51** | **1.25** | **0.09** | **4.95** | **23.33** |

**Table 4.** Quantitative ablation study on the PET-MRI dataset using regular (Reg) and depthwise convolution (D/w). The best values are highlighted in bold.

### 4.4.5 Failure Cases

We observe several failure cases on the IVF task for the MSRS dataset, as shown in Figure 9, particularly in scenarios with large brightness differences between the two source images or under adverse weather conditions such as rain and fog. These failures are likely due to the limited number of such challenging samples in the training dataset.

## 5 Conclusions

Existing image fusion models often achieve strong performance through increased model complexity. However, in the context of image fusion, where inputs representing the same scene share inherent correlation, much of this complexity may be dedicated to processing redundant information. We argue that robust fusion can instead be achieved through better-regulated attention, rather than increased model complexity. In this paper, we propose a novel image fusion model based on an encoder-decoder framework that integrates two main features: CNN blocks with a dedicated attention module to better focus on discriminative features, and a Swin Transformer-based fusion block to capture complementary information between different inputs more effectively. Our design enables robust performance at reduced computational cost. We further evaluate the fused images on downstream tasks, including semantic segmentation on the MSRS dataset, demonstrating that the fused outputs enhance task-specific performance. Extensive experiments across multiple image fusion tasks using several benchmark datasets show that our proposed model consistently performs competitively with state-of-the-art methods while maintaining significantly lower model complexity. Cross-validation highlights the strong generalization capability of our model, and ablation studies further confirm the effectiveness of our key design choices.

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

# 6 Appendix

For validating the generalization ability of our model, we compared the qualitative performance similarly with other state-of-the-art methods on the Lytro dataset and MEFB dataset (we train on the MFI-WHU dataset and test on the Lytro dataset; we train on SICE, and test on the MEFB dataset) as shown in Figure 10 and Figure 11. The fused images produced by our approach remain competitive with other state-of-the-art methods, displaying high perceptual quality with fewer visual artifacts and less exposure or brightness bias.

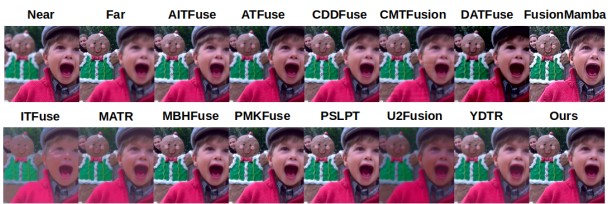

**Figure 10.** Qualitative comparison on the Lytro dataset (MFF Task) with other state-of-the-art methods.

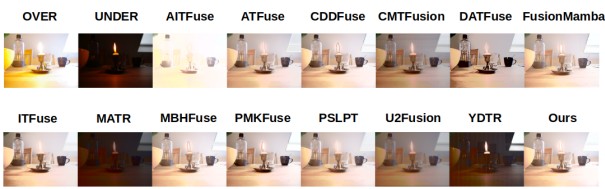

**Figure 11.** Qualitative comparison on the MEFB dataset (MEF Task) with other state-of-the-art methods.

Furthermore, qualitative segmentation results on the MSRS dataset using DeepLabV3 with pre-

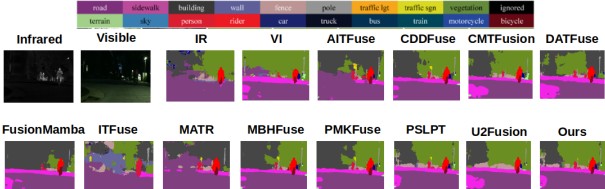

**Figure 12.** Qualitative comparison on the MSRS dataset with state-of-the-art methods for the downstream semantic segmentation task, using DeepLabV3 with pre-trained weights from the Cityscapes dataset.

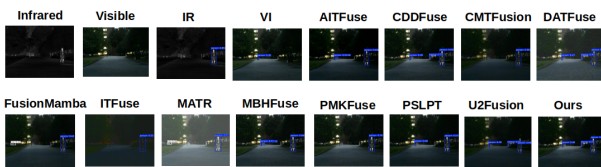

**Figure 13.** Qualitative comparison on the MSRS dataset with state-of-the-art methods for the downstream object detection task, using YoloV5 with pre-trained weights from the COCO dataset.

trained weights from the Cityscapes dataset are presented in Figure 12.

We further evaluate the fused images on the object detection task using the MSRS dataset, employing YoloV5 with pre-trained weights from the COCO dataset. The qualitative comparison is shown in Figure 13.

