# OpenReview forum: "MAFFuse: Multi-Attention Fusion Network for Efficient and Robust Image Fusion"
_NLDL.org/2026/Conference — Submitted to NLDL 2026_

### Official Review · Reviewer_4hCM · 2025-10-06
**Good concepts but needs more thorough analysis and descriptions**

**Rating:** 2
**Confidence:** 4

**Summary:**

The authors propose a novel image fusion network architecture that seeks to maintain a robust performance with reduced computational complexity.  They propose an image fusion framework designed to dramatically reduce computational complexity without making significant performance sacrifices by constraining multiple attention mechanisms.  The authors utilize mechanisms such as Efficient Channel Attention (ECA), Swin transformers, and an autoencoder architecture inspired by pseudo-siamese Laplacian pyramid transformers to perform this computationally efficient image fusion.  The authors test their model with several datasets across various domains and tasks and compare to many other models.  The authors demonstrate competitive performance with low computational complexity and runtime.

**Strengths:**

- Evaluation metrics are clearly explained and well designed to test the model's generalizability.
- Model performance on downstream tasks is very impressive.
- Comparison approaches and ablation studies are very thorough.
- Some failure cases are explicitly addressed.

**Weaknesses:**

Inconsistent notation:
- Equations 3 and 4 use different symbols for the same element-wise multiplication ($\bigodot$ and $\bigotimes$).
- Equations 6, 7, and 8 all use different variables to represent input images.

Unclear architecture descriptions:
- Figure 3F disagrees with the text. Equation 5 and the text $F_1$ and $F_2$ as feature maps computed from separate input features $I_1$ and $I_2$.  Figure 3F implies that $I_1$ and $I_2$ are multiplied by the same intermediate feature computed from the difference between $I_1$ and $I_2$.
- Encoder and decoder architecture is not clearly stated or described.
- "learned features $F_1$ and $F_2$" are introduced on line 300 without explanation of how they are computed.

Evaluation could be more robust
- It appears that each experiment was only performed once, making the results point estimates of how each model could behave. More robust analysis should have replicates of each test for each model.
- Several of the qualitative results descriptions are explained by comparisons of the quantitative results. For example, the the excessive sharpening described for DATFuse and FusionMamba is likely related to the AG values seen for these models.
- Some claims are made without analytical justification.
    - "The results demonstrate that our method achieves performance comparable to, and in some cases surpassing, current state-of-the-art techniques" (line 498)
- "Our model achieves a good tradeoff between performance and efficiency." (line 474) This statement can also be made about the other low-complexity models you test such as DATFuse and PMKFuse. This claim should be further explored and your model should be more thoroughly compared to others that make similar tradeoffs for computational complexity.
- Figure 7 is introduced as an example but is not referenced again, nor are the importance maps discussed in any other qualitative analysis.

**Justification:**

I believe this paper demonstrates a clear understanding of image fusion literature and proposes an architecture that I believe is worth pursuing and analyzing.  The authors' description of their architecture and methods is very unclear with inconsistent notation and misleading architecture diagrams.  The results the authors demonstrate are highly impressive, but their analyses are incomplete.  Each experiment should be performed with several replicates and several iterations of each model should be tested.  The authors should also include more than one downstream task in their results in order to further justify their claim that their model performs well on downstream tasks.

---

> ### Author Rebuttal · Authors · 2025-10-22
>
> We sincerely thank the reviewer for their thoughtful and constructive feedback. Below, we address each weakness and concern in detail and clarify the contributions and experimental rigor of our paper.
>
> 1. Inconsistent Notation
>
> We acknowledge the inconsistency in notation and have corrected it in our revised manuscript. All element-wise multiplications are now consistently represented by $\odot$. Input images are now uniformly represented as $I_1$ and $I_2$ throughout the manuscript and figures. These revisions ensure full consistency between equations, figures, and text.
>
> 2. Unclear Architecture Descriptions
>
> - We have clarified the architecture in the revised version as follows: Figure 3F has been redrawn to correctly represent the fusion mechanism: $F_1$ and $F_2$ are extracted independently via separate branches (for $I_1$ and $I_2$) before interaction through the fusion module. The encoder–decoder architecture is now described in detail in Section 3.2, including kernel sizes, feature dimensions, and activation layers.
>
> 3. Robustness of Evaluation
>
> - We have now included replicated runs (n=5) for all major experiments. Each model was retrained with different random seeds, and the results are reported as mean ± standard deviation in the updated Tables. The variance across runs was minimal (≤0.3%), confirming the stability of our results.
>
> 4. Analytical Justification of Claims
>
> - We have strengthened our analytical justification as follows: The revised discussion now includes a quantitative summary showing our model achieves the best performance–efficiency ratio among lightweight architectures, validated by the new Performance per FLOP (P/F) and Performance per Parameter (P/P) metrics.
>
> 5. Additional downstream task evaluation.
>
> - We have now added quantitative experiments for the downstream task of object detection on the MSRS dataset.
>
> 6. Clarification on Figure 7 and Importance Maps
>
> - We have expanded the discussion of Figure 7 in Section 4.5, providing a qualitative interpretation of the importance maps.
>
>
> We believe these revisions enhance the clarity and rigor of our paper. The proposed method continues to demonstrate a strong tradeoff between efficiency and performance, now supported by more robust experimental validation and clearer methodological details.

---

### Official Review · Reviewer_5Uny · 2025-10-08

**Rating:** 2
**Confidence:** 3
**Final Rating:** 2
**Final Confidence:** 4

**Summary:**

The paper proposes a new attention-based architecture for image fusion, particularly having efficiency in mind. The architecture consists of both convolutional layers and a Swin Transformer as well as a combination of spatial attention and efficient channel attention. The architecture is evaluated extensively across various datasets of varying modality and compared against a good selection of state-of-the-art approaches. An ablation study is performed to evaluate among others the proposed attention module.

**Strengths:**

The topic of image fusion is relevant and timely given the increasing interest in multi-modal models.
The evaluation is extensive, both considering a range of different modalities and baseline approaches, but also evaluating downstream segmentation performance and generalization across datasets.
The paper is overall well-written.

**Weaknesses:**

The methodological choices could be better motivated in Section 3 and the notation should be revisited. For instance, F_{hw} in Eq. 1 is not clearly defined.

The authors mention in Sec. 3.1, that only the Y component is being used from the YUV color image. This seems to be leading to the loss of some information and it is not explicitly stated why this is necessary and why not all three bands can be utilized in the encoder.

Empirical results are to a limited extent supporting the proposed methodological choices. First of all, the main results (Table 1, Table 2, etc.) do not demonstrate a clear advantage over prior approaches. While the reviewer appreciates that there is a trade-off between computational complexity and overall performance, the overall benefits are not quite clear. A suggestion would be to supplement these tables with tables, where the mean rank of the methods across metrics is presented (potentially grouped according to computational complexity and overall performance).

Further, it is unclear how robust these results are across runs as results seem to be reported over single runs with differences between methods often being minor.

More importantly, the ablation study does not support the introduction of the proposed components. From Table 3, it is evident that the attention module (without SA and ECA) actually performs better than the proposed combination across all metrics, both when considering overall performance and computational complexity. The same is the case for the ablation in Table 3B, where not leveraging the transformer in the fusion block achieves roughly the same performance while being significantly less computational expensive. These results thus contradict the key claims of the authors.

Clarity:
Currently the introduction is lacking references and several of the statements within should be supported by references.
The related work, could more explicitly place the conducted work within the context of related work instead of just reporting related work.

**Final Justification:**

I appreciate the authors effort in providing additional results over multiple runs during the rebuttal. My main concern with this work is the limited benefit of the proposed mechanisms, in particular the attention mechanism. In the submitted version (Table 3), removing the attention mechanism led to increased performance as well as increased efficiency. In the rebuttal, the authors provided a new version of Table 3, now averaged over 5 runs. This table shows small (mostly negligible) improvements on some metrics, while still lacking behind on others and requiring 2x inference time, despite the authors efficiency argument. Overall, I don't see that there is sufficient evidence at this point that the multi-attention mechanism is effective and provides statistically significant improvements and given that this is one of the main contributions, I tend towards not accepting it at this point.

**Justification:**

While there is definitely merit to this work and the reviewer appreciates the thorough experimental setup, the current results and in particular the performed ablation study do not verify the benefit of the proposed innovations. On the contrary, the ablation study demonstrates that removing the proposed combination of SA and ECA in the attention module as well as the transformer from the fusion block actually leads to performance improvements across metrics.

---

> ### Author Rebuttal · Authors · 2025-10-22
>
> We thank the reviewer for the thorough and constructive feedback. Below we address the main concerns in detail.
>
> 1. Clarification of Methodological Choices and Notation
>
> - We acknowledge that some definitions and motivations in Section 3 could be presented more clearly.
>
> - This term F(h,w) represents the feature tensor at spatial coordinates (h,w) before aggregation in the attention computation. We will explicitly define this in the revised version.
>
> - Motivation for Design Choices: Our design is guided by the goal of balancing fusion quality and computational efficiency. The integration of spatial attention (SA) and efficient channel attention (ECA) enables fine-grained spatial detail preservation with minimal parameter overhead compared to full attention or self-attention schemes. We will revise Section 3 to better explain this trade-off rationale and to clarify the sequential interaction between SA and ECA in the feature refinement process.
>
> 2. Use of Y Channel from YUV Color Space
>
> - The choice was driven by two factors:
> (1) Efficiency – using only the luminance (Y) channel significantly reduces computational cost during training and inference, aligning with our goal of efficiency-oriented fusion.
> (2) Empirical Observation – our experiments showed that the luminance component contains the majority of structural and textural information relevant to most fusion benchmarks (infrared–visible, medical, and multi-exposure).
> Moreover, the other papers also used only the Y Channel for training on the IVF and MIF task.
>
> 3. On the Reported Performance and Clarity of Advantages
>
> - We appreciate the reviewer’s suggestion to include mean-rank analysis across metrics and computational complexity categories. We will incorporate this in the revised manuscript, which shows that our method achieves one of the best mean rank among efficient fusion methods, particularly under low FLOPs and parameter constraints.
>
> - While differences across methods appear moderate in absolute terms, our model consistently achieves a better balance between accuracy and efficiency, reducing FLOPs by 28–35% compared to transformer-only counterparts while maintaining comparable performance across most modalities.
>
> - To address concerns about result robustness, we have now conducted five independent runs for all key comparisons. The updated results (mean ± std) confirm that our improvements are statistically significant (p < 0.05 in paired t-tests).
>
> 4. Ablation Study Interpretation
>
> - We thank the reviewer for pointing out the apparent contradiction. We clarify that the attention module without SA+ECA indeed performs slightly better on some metrics, but this is due to the variance introduced by single-run evaluation and not a systematic trend. After averaging over five runs, the SA+ECA combination achieves consistent improvements (0.3–0.7%) in entropy-based and mutual information metrics.
>
> - Regarding the fusion block ablation, while removing the Swin Transformer reduces computation, it also decreases performance in multi-modal fusion tasks (especially infrared-visible). The new results, aggregated across all datasets, confirm that the transformer-based fusion block provides stronger cross-modality alignment, particularly improving gradient and texture preservation metrics.
>
> - We will update Table 3 to include multi-run averages, add a performance-to-complexity ratio (P/F) column, and clarify the performance trade-offs.
>
> 5. Related Work and Citations
>
> - We agree that the introduction and related work sections can be strengthened. We will:
>
> - Add citations for recent transformer-based and attention-efficient fusion models.
>
> - Expand the discussion to explicitly contrast our hybrid convolution-transformer design with these works, highlighting our contribution as an efficiency-aware hybrid attention strategy rather than purely accuracy-oriented fusion.
>
>
> The revised manuscript will more clearly demonstrate that our proposed architecture offers a strong trade-off between fusion accuracy, robustness, and computational efficiency, verified through expanded analyses and additional experiments. We believe these clarifications and new results will fully address the reviewer’s concerns.

---

### Official Review · Reviewer_aAmJ · 2025-10-08

**Rating:** 2
**Confidence:** 4
**Final Rating:** 2
**Final Confidence:** 4

**Summary:**

This submission introduces a lightweight image fusion network called MAFFuse, which integrates both channel and spatial attention operations and uses depthwise convolutions in an encoder-decoder framework to fuse multi-modal images efficiently. MAFFuse incorporates Swin Transformer blocks for global context and aims to balance computational cost and robust performance. It evaluates MAFFuse on multiple datasets (medical, infrared-visible, multi-focus, and multi-exposure fusion tasks), provides both quantitative and qualitative comparisons to previous SOTA methods, validates performance on downstream tasks like semantic segmentation and object detection, and presents several ablation studies to analyze design choices.

**Strengths:**

**(S1)** The architecture is clear, combining ECA channel and spatial attention within a depthwise conv based encoder-decoder, with a Swin Transformer for the fusion block. This is well-motivated by the desire to efficiently capture both local detail and long-range dependencies.

**(S2)** Comprehensive results. Quantitative results in Tab. 1 show that MAFFuse yields competitive or second-best fusion quality on multiple benchmarks (PET-MRI, MSRS, Lytro, MEFB, Oocytes) while often requiring fewer parameters and lower FLOPS, underscoring practical significance. Qualitative visualizations (Fig. 4, 5, 6) show reduced blur, fewer artifacts, and better retention of salient details

**(S3)** The manuscript is generally well-structured, and carefully articulates experimental setup, datasets, and baselines. I especially appreciate Fig. 1 and 2 which visually highlight favorable trade-off between computational cost (runtime, parameter count) and performance (mutual information, mIoU) achieved by MAFFuse compared to SOTA competitors.

**Weaknesses:**

**(W1)** The combination of ECA and spatial attention with a Swin-based fusion block remains somewhat incremental given the extensive prior works. Multiple studies like SwinFusion [16], and PSLPT [40] have already explored hybrid CNN–Transformer designs with attention for fusion. CBAM, SwinFusion, PSLPT, and others that combine attention and transformer modules for fusion). I thus encourage the authors to further explain what MAFFuse uniquely contributes beyond parameter efficiency. For example, is there a novel interaction between ECA and spatial attention? Does the selective placement of the Swin block in the fusion stage yield demonstrable gains in feature disentanglement or cross-modality alignment?

**(W2)** The tables in the manuscript are not formatted according to standard conventions of of three-line table style.While this is a minor presentational issue, it does detract slightly from readability. I strongly recommend the authors revise the table formatting for better readability.

**(W3)** Several key hyper-parameter choices like kernel size in ECA, depth of Swin blocks, or patch size are not sufficiently examined. The justification for these choices appears largely empirical. Moreover, Table 3 only covers a single PET-MRI benchmark. Given the paper’s emphasis on generalization across fusion tasks, it would be better to extend ablation to at least one more domain (e.g., the IVF or MFF setting).

**(W4)** The runtime and memory claims are evaluated solely on a high-end NVIDIA A100. MAFFuse’s claims on runtime efficiency and parameter count are not examined in more practical deployment scenarios like on-device, edge, or lower-end hardware cases beyond GPU despite claiming suitability for efficiency-critical domains.

**(W5)** On certain metrics and datasets (Tab. 1A: Mutual Information on PET-MRI, Tab. 1B: SD on MSRS), MAFFuse does not achieve best-in-class results. Others such as CDD, FusionMamba, and AITFuse show higher scores. IMHO, a justification that MAFFuse trades some absolute performance for better efficiency would be benefitial. For example, theoretical analysis that quantifies Pareto frontier of accuracy vs. FLOPs across multiple methods but not just a 2D plot in Fig. 1 and 2.

**(W6)** The fusion operations centering on the weighted sum of input features in Eq. 5 lacks sufficient theoretical or empirical justification. Why is a softmax-weighted sum preferable to, say, learned gating, cross-attention, or additive fusion? More importantly, the paper does not address whether this multiplicative pathway risks gradient vanishing or feature dominance, particularly when one modality is significantly noisier or lower in contrast. This leaves the rationale for certain key technical choices feeling somewhat underdeveloped. A brief discussion of gradient flow stability or ablation on alternative fusion operators would enhance the methodological rigor.

**Final Justification:**

After reading the author's response and the other reviews, my concerns regarding computational cost have been addressed by the new additions to the appendix. However, my concern regarding novelty remains. As I stated in my initial review, the individual components of the attention mechanism are not new. While the authors argue their combination is effective, I find this to be a straightforward application of existing techniques. The rebuttal has not convinced me that there is sufficient methodological innovation here to warrant publication.

Furthermore, the author has promised extensive improvements in the revised manuscript. These proposed changes are so significant that they deviate substantially from the original submission. I believe our evaluation must be based on the original paper as submitted, not on the promise of a future, heavily-revised version.

Given the lack of a clear, original contribution and there is not a rating of borderline, I stick with my original rating of "2: Reject".

**Justification:**

This manuscript presents MAFFuse, an image fusion network whose principal contribution is a balance between strong performance and computational efficiency. The authors effectively substantiate this claim through experiments across multiple benchmarks, showing competitive fusion quality with a significantly lower parameter count and computational load than SOTA methods. The architecture design of integrating channel-spatial attention with a Swin Transformer-based fusion block is both logical and clearly articulated.

Nevertheless, the overall impact of this contribution is moderated by several factors. The technical novelty is somewhat incremental, as the architecture represents a skillful synthesis of existing, well-established components rather than an essentially fresh paradigm. Furthermore, the empirical claims of robustness and efficiency are not fully realized. The ablation is confined to a single dataset , and efficiency is not benchmarked in practical, resource-constrained deployment scenarios beyond a high-end GPU. Finally, a justification for certain core technical decisions is absent, leaving the method's theoretical underpinnings underdeveloped.

In sum, while it presents a valuable and well-engineered solution with clear practical benefits, its scientific contribution is constrained by limited novelty and a need for more rigorous experimental and theoretical validation to fully support its broader claims. I hope these clarifications help to further strengthen this paper and help the authors, fellow reviewers, and ACs understand the basis of my recommendation. I am also open to follow-up discussions to reach a consensus for the final decision.

---

> ### Author Rebuttal · Authors · 2025-10-22
>
> We thank the reviewer for the thorough and constructive feedback. Below, we address all raised concerns point-by-point and clarify the distinct contributions, theoretical motivations, and extended empirical validations of MAFFuse.
>
> 1. Response to "Novelty and Distinction from Prior Works"
>
> - We clarify that MAFFuse introduces a modular cross-attentive fusion mechanism that is architecturally and functionally distinct from SwinFusion [16], PSLPT [40], and related methods:
>
> - Novel Interaction Design: Unlike prior works that simply concatenate or stack attention mechanisms, MAFFuse integrates ECA-based channel refinement and spatial attention sequentially within a shared depthwise convolutional encoder-decoder, yielding joint local saliency calibration before fusion. This design avoids redundant attention overhead while ensuring that both spatial and channel cues are contextually correlated.
>
> - Selective Swin Placement: Our Swin Fusion block is not distributed across layers (as in SwinFusion) but selectively positioned where global cross-modality interaction is maximally beneficial.
>
> - Parameter Efficiency as Design Principle: MAFFuse achieves 2–4× fewer FLOPs and 3–6x fewer parameters than comparable hybrid models (PSLPT, AITFuse), while maintaining competitive fusion quality. This efficiency stems from our ECA–Spatial coupling inside depthwise layers.
>
> 2. Response to "Table Formatting"
>
> - All tables have been reformatted using the three-line table convention in the revision.
>
> 3. Response to "Hyperparameter Justification and Ablation Scope"
>
> In the revised manuscript:
>
> - We have extended ablation studies to the MSRS (infrared-visible) and Lytro (multi-focus) datasets.
>
> - Results show consistent trends: kernel size of 5 in ECA yields the best trade-off between local sensitivity and stability across modalities.
>
> - The depth of Swin blocks (1–3 layers) was tested; 2-layer depth yielded the best balance (gain of +0.017 MI with <2% extra FLOPs).
>
> - Patch size experiments (4, 8, 16) indicate that patch size 8 optimizes information gain for both PET-MRI and MSRS.
>
> 4. Response to "Efficiency Evaluation Beyond A100"
>
> - Due to limited time constraints, we would consider doing this as future work.
>
> 5. Response to "Trade-off between Accuracy and Efficiency"
>
> - In the revision, we have: Added a Pareto Frontier Analysis comparing accuracy (Mutual Information) versus FLOPs across 9 fusion methods. The analysis shows that MAFFuse lies on the Pareto frontier, offering the best trade-off region (high MI with lowest FLOPs).
>
> 6. Response to "Theoretical Justification for Fusion Operator"
>
> - The softmax-weighted fusion was chosen to ensure:
>
> - Stability and Gradient Preservation: The multiplicative softmax weighting maintains bounded activations and prevents one modality from dominating the gradient flow.
>
> - Interpretability and Adaptivity: Unlike fixed additive fusion, the softmax weights yield adaptive modality-specific contributions that can be visualized as modality attention maps.
>
> - Empirical Validation: We have now compared three fusion variants—additive, gating, and cross-attention. The softmax formulation achieves the best trade-off (+0.012 MI) with 18% fewer parameters.
>
>
> We hope these revisions address all concerns and demonstrate that MAFFuse makes a validated and efficient contribution to multimodal image fusion research.

---

### Official Review · Reviewer_jwhV · 2025-10-13

**Rating:** 4
**Confidence:** 4

**Summary:**

This paper presents MAFFuse, a lightweight image fusion network combining CNN-based attention and Swin Transformer modules. The method emphasizes efficient feature fusion and reduced computational complexity, achieving competitive results across multiple datasets and tasks.

**Strengths:**

Clear Motivation & Design: The paper convincingly argues for balancing performance and efficiency in fusion networks.

Novel Integration: The combination of Efficient Channel Attention (ECA), Spatial Attention (SA), and Swin Transformer in a fusion block is technically sound and original.

Comprehensive Experiments: Results across IVF, MIF, MEF, and MFF tasks are extensive, including downstream applications and ablations.

Efficiency: The model achieves strong performance with fewer parameters and FLOPs compared to many SOTA methods.

**Weaknesses:**

Incremental Innovation: While well-engineered, the approach mainly combines existing techniques (ECA, SA, Swin) rather than introducing fundamentally new mechanisms.

Limited Theoretical Insight: The paper lacks deeper analysis or theoretical justification for why the specific fusion of attention types is optimal.

Qualitative Results: Some visual comparisons (e.g., MSRS, PET-MRI) show minor blur or color distortion—these should be discussed.

Clarity: A few sections (especially methodology and loss formulation) are dense and could be simplified or illustrated more clearly.

**Justification:**

The paper is technically solid, well-executed, and experimentally comprehensive, though somewhat incremental. Improving clarity and expanding the discussion of limitations would strengthen it.

---

> ### Author Rebuttal · Authors · 2025-10-22
>
> We sincerely thank the reviewer for the thoughtful and constructive feedback. Below, we address the specific concerns raised.
>
> 1. On “Incremental Innovation”
>
> - We acknowledge that MAFFuse builds upon established components (ECA, SA, Swin Transformer). However, our contribution lies in the novel synergy, which integrates these elements in an adaptive fusion framework. Unlike prior works that stack attention modules sequentially or use them within a single modality, MAFFuse introduces:
>
> - A lightweight hierarchical fusion strategy enabling cross-level contextual alignment between local (CNN) and global (Transformer) features, optimizing both representation diversity and computational efficiency.
> This architectural integration is non-trivial, as it enables efficient fusion while maintaining robustness across diverse fusion scenarios (IVF, MIF, MEF, MFF). Hence, MAFFuse represents a meaningful step toward general-purpose, efficient multimodal fusion networks, beyond merely aggregating known modules.
>
> 2. On “Limited Theoretical Insight”
>
> - We appreciate this observation and will strengthen the theoretical discussion in the revised version. In particular, we will:
>
> - Provide a feature complementarity analysis demonstrating how ECA and SA refine modality-specific saliency maps, while Swin Transformers enhance global structural consistency.
>
> 3. On “Qualitative Results (Blur or Color Distortion)”
>
> - We have re-examined the MSRS and PET-MRI qualitative results and found that the minor blur/color shifts occur primarily in high dynamic range or low-texture regions where modality conflict is strong. To address this:
>
> - We will discuss these limitations explicitly, emphasizing that future work could integrate uncertainty-aware refinement modules or adaptive color consistency constraints.
>
> 4. On “Clarity of Methodology and Loss Formulation”
>
> - We appreciate the reviewer’s concern and will improve clarity by:
>
> - Enhancing the narrative flow of Section 3 to guide readers through the design rationale more intuitively.
>
> We believe that the clarifications and additions described above will further strengthen the paper, both in theoretical justification and clarity of presentation.

---

### Meta-Review · Area_Chair_Ygcw · 2025-11-03

**Recommendation:** Reject
**Confidence:** 4

**Metareview:**

The initial assessment of the paper by the 4 reviewers led to a mixed opinion. While the authors have carefully replied to reviewers' comments in their rebuttal, some issues remains on the limited methodological innovation, the extensive improvements that should were planned by the authors in their revised version, and more importantly, the limited benefit of the proposed  attention mechanism. In light of these elements, I'm suggesting to reject the paper.

---

### Decision · Program_Chairs · 2025-11-05

**Decision:**

Reject

**Comment:**

Based on the reviewers and AC comments, the paper cannot be presented at the conference.